# Optimizing Learning for Robust Hyperbolic Deep Learning in Computer Vision

## Abstract

Hyperbolic deep learning has become a growing research direction in computer vision for the unique properties afforded by the alternate embedding space. The negative curvature and exponentially growing distance metric provide a natural framework for capturing hierarchical relationships between datapoints and allowing for finer separability between their embeddings. However, these methods are still computationally expensive and prone to instability, especially when attempting to learn the negative curvature that best suits the task and the data. Current Riemannian optimizers do not account for changes in the manifold which greatly harms performance and forces lower learning rates to minimize projection errors. Our paper focuses on curvature learning by introducing an improved schema for popular learning algorithms and providing a novel normalization approach to constrain embeddings within the variable representative radius of the manifold. Additionally, we introduce a novel formulation for Riemannian AdamW, and alternative hybrid encoder techniques and foundational formulations for current convolutional hyperbolic operations, greatly reducing the computational penalty of the hyperbolic embedding space. Our approach demonstrates consistent performance improvements across both direct classification and hierarchical metric learning tasks while allowing for larger hyperbolic models.

## 1 Introduction

With the recent rise in the use of hyperbolic manifolds for deep representation learning, there is a growing need for efficient, flexible components that can fully exploit these spaces without sacrificing stability. This has led researchers to focus on two main derivations of hyperbolic space: the Poincaré manifold and the hyperboloid. The Poincaré ball, equipped with a gyrovector space, supports various well-defined operations, including generalized vector addition and multiplication, but it suffers from significant stability issues. On the other hand, the hyperboloid, or Lorentz space, lacks these operations but offers much better operation stability, as demonstrated in the study by Mishne et al. (2022).

To address this gap, previous works have sought to provide Lorentzian definitions for common deep learning operations such as the feed-forward layer (Chen et al., 2022; Dai et al., 2021; Ganea et al., 2018), convolutional layer (Chen et al., 2022; Qu & Zou, 2023; Dai et al., 2021), and MLRs (Bdeir et al., 2024). This increased focus on hyperbolic modeling has led to its gradual integration into computer vision architectures, as detailed in the survey by Mettes et al. (2023). Specifically, the hyperboloid model has been employed as a sampling space for VAEs (Nagano et al., 2019), a decoder space for vision tasks in hybrid settings (Guo et al., 2022; Liu et al., 2020; Khrulkov et al., 2020; Qu & Zou, 2022), and ultimately for fully hyperbolic Lorentzian vision encoders (Bdeir et al., 2024) simultaneously with its Poincaré counterpart (van Spengler et al., 2023).

This paper furthers the development of hyperbolic learning for vision tasks, specifically for the Lorentz manifold. Our primary focus is on the challenge of learning the manifold's negative curvature. The driving principle behind this, is that the model embeddings may exhibit varying degrees of hyperbolicity depending on the innate hierarchies in the datapoints themselves, the problem task that is being considered, and the specific locations of hyperbolic operation integrations. To accomodate for this, we can adjust the embedding space's hyperbolic metric to be less or more Euclidean which

accounts for the modeling requirements. We also build on the idea of separate manifolds for separate main blocks in the architecture further increasing representative flexibility.

We also recognize that despite recent advances, Lorentz models continue to struggle with issues of high computational costs. We attempt to isolate and alleviate the main factors leading to numerical inaccuracies and computational overhead overall, and more particularly when modeling data in higher-dimensional embedding spaces and when learning the curvatures. Our contributions can then be summed up as:

1. We propose a formulation for Riemannian AdamW and an alternative schema for Riemannian optimizers that accounts for manifold curvature learning.

2. We propose the use of our maximum distance rescaling function to restrain hyperbolic vectors within the representative radius of accuracy afforded by the number precision, even allowing for fp16 precision.

3. We provide a more efficient convolutional layer approach that is able to leverage the highly optimized existing implementations.

4. We empirically show the effectiveness of combining these approaches using classical image classification tasks and hierarchical metric learning problems.

## 2    RELATED WORK

**Hyperbolic Embeddings in Computer Vision**    With the success of employing hyperbolic manifolds in NLP models (Zhu et al., 2021; Dhingra et al., 2018; Tifrea et al., 2018) hyperbolic embeddings have extended to the computer vision domain. Initially, many of the works relied on a hybrid architecture, utilizing Euclidean encoders and hyperbolic decoders (Mettes et al., 2023). This was mainly due to the high computational cost of hyperbolic operations in the encoder, as well as the lack of well-defined alternatives for Euclidean operations. However, this trend has begun to shift towards the utilization of fully hyperbolic encoders as can be seen in the hyperbolic Resnets by Bdeir et al. (2024) and van Spengler et al. (2023). Both works offer hyperbolic definitions for 2D convolutional layer, batch normalization layer, and an MLR for the final classification head. Bdeir et al. (2024) even attempt to hybridize the encoder by employing the Lorentz manifold in blocks that exhibit higher output hyperbolicity. While this has led to notable performance improvements, both models suffer from upscaling issues. Attempting to replicate these approaches for larger datasets or bigger architectures becomes much less feasible in terms of time and memory requirements. Instead, our approach places higher focus on efficient components to leverage the beneficial hyperbolic properties of the model while minimizing the memory and computational footprint.

**Curvature Learning**    Previous work in hyperbolic spaces has explored various approaches to curvature learning. In their studies, Gu et al. (2018) and Giovanni et al. (2022) achieve this by using a radial parametrization that implicitly models variable curvature embeddings under an explicitly defined, fixed 1-curve manifold. This method enables them to simulate K-curve hyperbolic and spherical operations under constant curvature for the mixed-curve manifold specifically, a combination of the Euclidean, spherical, and Poincaré manifold. Other approaches, such as the one by Kochurov et al. (2020), simply set the curvature to a learnable parameter but do not account for the manifold changes in the Riemannian optimizers. This leads to hyperbolic vectors being updated with mismatched curvatures and others being inaccurately reprojected, resulting in instability and accuracy degradation. Additionally, some methods, like the one by Kim et al. (2023), store all manifold parameters as Euclidean vectors and project them before use. While this approach partially mitigates the issue of mismatched curvature operations, it remains less accurate and more computationally expensive. In comparison, our proposed optimization schema maintains the parameters on the manifold and optimizes them directly by performing the necessary operations to transition between the variable curvature spaces.

**Metric Learning**    Metric learning relies on the concept of structuring the distribution in the embedding space so that related data points are positioned closer together, while less related points are placed further apart. To facilitate this process, numerous studies have introduced additional loss functions that explicitly encourage this behavior. Contrastive losses, for instance, operate on pairs

of data points and propose a penalty that is proportional to the distances between negative pairs and inversely proportional to the distance between positive pairs (Chopra et al., 2005). Triplet loss extends this idea by considering sets of three points: an anchor, a positive sample, and a negative sample (Wang et al., 2014). Instead of changing the distances between points absolutely, it ensures that the distance between the anchor and the positive sample is less than the distance between the anchor and the negative sample, plus a margin, thus enforcing a relational criterion.

These approaches have also been adapted to hierarchical problem settings under hyperbolic manifolds (Yang et al., 2022; Kim et al., 2023). Notably, Kim et al. (2023) developed a method for learning continuous hierarchical representations using a deep learning, data-mining-like approach that relies on the innate relationships of the embeddings rather than their labels. They employ a proxy-based method that models the data on the Poincaré ball, facilitating a more natural extension to hierarchical tasks. Building on this, we extend the approach by modeling the loss function in the Lorentz manifold and incorporating a learnable curvature to better handle data with varying levels of hierarchy.

## 3 METHODOLOGY

Hyperbolic embeddings offer significant advantages for computer vision tasks that benefit from hierarchical structure modeling, but their effectiveness is hindered by practical challenges, including mathematical inconsistencies in curvature learning, the absence of a Riemannian AdamW optimizer, stability issues with float32 precision, and high computational costs from inefficient convolutions.

To address these issues, the following sections introduce four key advancements: a more stable curvature learning method, a Riemannian AdamW optimizer, a normalization scheme for improved float32 precision stability, and efficient hyperbolic convolutions to reduce memory and computational costs.

### 3.1 BACKGROUND

The hyperbolic space is a Riemannian manifold with a constant negative sectional curvature $c < 0$. There are many conformal models of hyperbolic space but we focus our work on the hyperboloid, or Lorentz manifold. The n-dimensional Lorentz model $\mathbb{L}_K^n = (\mathcal{L}^n, \mathfrak{g}_{\boldsymbol{x}}^K)$ is defined with $\mathcal{L}^n := \{\boldsymbol{x} \in \mathbb{R}^{n+1} \mid \langle \boldsymbol{x}, \boldsymbol{x} \rangle_{\mathcal{L}} = \frac{-1}{K}, \ x_t > 0\}$ where $\frac{-1}{K} = c$, and with the Riemannian metric $\mathfrak{g}_{\boldsymbol{x}}^K = \mathrm{diag}(-1, 1, \ldots, 1)$. This models the upper sheet of a two-sheeted hyperboloid centered at $\overline{\boldsymbol{0}} = [\sqrt{K}, 0, \cdots, 0]^T$. We inherit the terminology of special relativity and refer to the first dimension of a Lorentzian vector as the time component $x_t$ and the remainder of the vector, the space dimension $\boldsymbol{x_s}$. The Lorentzian inner product then becomes $\langle \boldsymbol{x}, \boldsymbol{y} \rangle_{\mathbb{L}} := -x_t y_t + \boldsymbol{x}_s^T \boldsymbol{y}_s = \boldsymbol{x}^T \mathrm{diag}(-1, 1, \cdots, 1)\boldsymbol{y}$. We now define the common hyperbolic operations in the Lorentz space.

**Distance** Distance in hyperbolic space is the magnitude of the geodesic forming the shortest path between two points. Let $\boldsymbol{x}, \boldsymbol{y} \in \mathbb{L}_K^n$, the distance between them is given by $d_{\mathbb{L}}(\boldsymbol{x}, \boldsymbol{y}) = \sqrt{K} \cosh^{-1}(\frac{-\langle \boldsymbol{x}, \boldsymbol{y} \rangle_{\mathbb{L}}}{K})$. We also define the square distance by Law et al. (2019) as $d_{\mathbb{L}}^2(\boldsymbol{x}, \boldsymbol{y}) = ||\boldsymbol{x} - \boldsymbol{y}||_{\mathbb{L}}^2 = -2K - 2\langle \boldsymbol{x}, \boldsymbol{y} \rangle_{\mathbb{L}}$.

**Exponential and Logarithmic Maps** Seeing as the Lorentz space is a Riemannian manifold, it is locally Euclidean. This can best be described through the tangent space $\mathbb{T}_{\boldsymbol{x}}\mathcal{M}$, a first-order approximation of the manifold at a given point $\boldsymbol{x}$. The exponential map, $\exp_{\boldsymbol{x}}^K(\boldsymbol{z}) : \mathbb{T}_{\boldsymbol{x}}\mathbb{L}_K^n \to \mathbb{L}_K^n$ is then the operation that maps a tangent vector $\mathbb{T}_{\boldsymbol{x}}\mathbb{L}_K^n$ onto the manifold through $\exp_{\boldsymbol{x}}^K(\boldsymbol{z}) = \cosh(\alpha)\boldsymbol{x} + \sinh(\alpha)\frac{\boldsymbol{z}}{\alpha}$, with $\alpha = \sqrt{1/K}||\boldsymbol{z}||_{\mathbb{L}}$, $||\boldsymbol{z}||_{\mathbb{L}} = \sqrt{\langle \boldsymbol{z}, \boldsymbol{z} \rangle_{\mathbb{L}}}$. The logarithmic map is the inverse of this mapping and can be described as $\log_{\boldsymbol{x}}^K(\boldsymbol{y}) = \frac{\cosh^{-1}(\beta)}{\sqrt{\beta^2 - 1}} \cdot (\boldsymbol{y} - \beta \boldsymbol{x})$, with $\beta = -\frac{1}{K}\langle \boldsymbol{x}, \boldsymbol{y} \rangle_{\mathbb{L}}$.

### 3.2 RIEMANNIAN OPTIMIZATION

**On the Stability of Curvature Learning** In the hyperbolic learning library GeoOpt, Kochurov et al. (2020) introduce the curvature of hyperbolic space as a learnable parameter. However, no subsequent

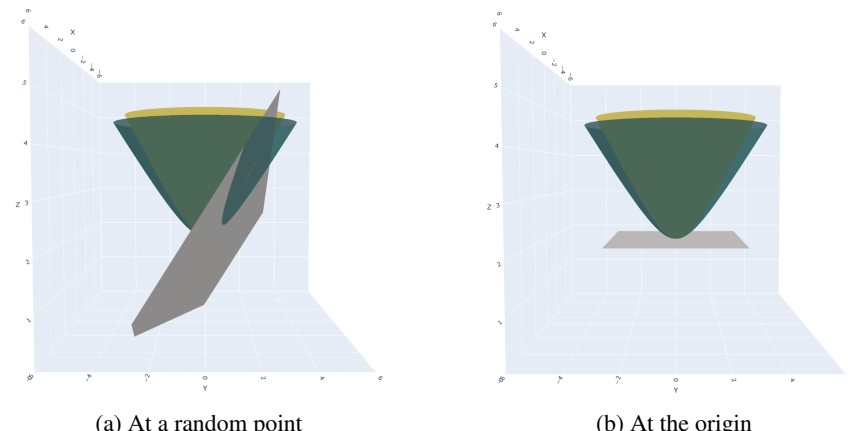

(a) At a random point          (b) At the origin

Figure 1: Tangent planes of a hyperboloid with curvature -1 relative to another hyperboloid with curvature -0.7. Tangential properties between manifolds are better respected at the origin where tangents remain parallel.

work has fully leveraged this feature, and our empirical testing reveals that this approach often leads to instability and performance decline. We attribute these issues to the simplistic implementation of the curvature updates, which neglects to update the hyperbolic operations after updating the curvature.

In the current Riemannian optimization approach, Euclidean gradients and their momentums are projected onto the Riemannian manifold and then used to update the hyperbolic parameters. These operations inherently depend on the curvature of the space. However, when the curvature is adjusted during learning, the projections, gradients, and hyperbolic parameters can become inconsistent, as they mix the projection and update operations from the new curvature with vectors defined under the previous curvature.

To illustrate this, Figure 1 shows two manifolds with different curvatures. It is evident that points defined on the first manifold do not lie on the second, and the tangent planes of the first manifold are not aligned with those of the second. Thus, using the new curvature operations on points from the old model is not mathematically sound practice. This inconsistency introduces instability, undermining the reliability of curvature learning and often leading to invalid values during the parameter updates. To address this, we propose a method that tracks both the old and new curvature values throughout updates. Specifically, all updates are applied under the old curvature, after which the hyperbolic vectors are mapped from the old manifold to the new one.

While GeoOpt does not address the optimization challenges involved in learning manifold curvature, it does implement an "N-stabilize step". This step aims to reduce instability by recalculating the time component of the hyperbolic parameters every N-steps to ensure they remain on the manifold. However, this approach fails to prevent invalid parameter values before the update steps and suffers from high inaccuracies due to the mathematically flawed parameter optimization.

Moreover, the time re-calculation projection used in the stabilize step alters the relative magnitudes of the hyperbolic vectors and gradients, and even the directions of the momentums, which can degrade performance and lead to training instability. Instead, we recommend projecting points onto the tangent space at the old origin using the logarithmic map and then projecting back after the curvature update. Gradients and their momentum can be similarly projected or parallel transported to the tangent space. Parallel transport is chosen here because it preserves the vectors' directions which is important for the optimization procedure.

Additionally, when the curvature changes, the tangent space at the origin remains relatively stable, as shown in Figure 1. Moving onto the new manifold is then a simple exponential map and parallel transport back operation. This preserves the vector norms and their hyperbolic distances to the origin since $D(\exp_{\overline{\mathbf{0}}}^{K_1}(\boldsymbol{y}), \overline{\mathbf{0}})^{K_1} = ||y|| = D(\exp_{\overline{\mathbf{0}}}^{K_2}(\boldsymbol{y}), \overline{\mathbf{0}})^{K_2}$ where $\boldsymbol{y} \in \mathcal{T}_{\overline{\mathbf{0}}}\mathcal{M}$. No changes are needed for the tangent vector during the move between tangent spaces of different curvatures since these spaces are parallel, making the transition a simple translation along the time axis.

It is important to emphasize that our proposed optimization scheme is compatible with existing curvature learning and meta-learning methods. Rather than being an alternative, it serves as a guideline for updating manifold parameters during curvature changes, aimed at maintaining learning stability throughout the process.

**Riemannian AdamW Optimizer** Recent works, particularly with transformers, commonly use the AdamW optimizer introduced by Loshchilov & Hutter (2019). As of current, there is no established Riemannian variant of this optimizer. We attempt to derive AdamW for the Lorentz manifold and argue a similar approach could be extended to the Poincaré ball. The main difference between AdamW and Adam is the direct weight regularization. This is more difficult to perform in the Lorentz space given the lack of an intuitive subtraction operation on the manifold. To resolve this, we reframe the parameter regularization in the original AdamW as a weighted centroid with the origin $O$

$$\boldsymbol{\theta_{t-1}} - \gamma\lambda\boldsymbol{\theta_{t-1}} = (1 - \gamma\lambda)\boldsymbol{\theta_{t-1}} + \gamma\lambda O$$

where $\gamma$ is the learning rate and $\lambda$ is the weight decay value. The centroid operation is well defined in the Lorentz space which now allows for a direct translation of AdamW. The regularization schema becomes:

$$\boldsymbol{\theta_t} = \begin{cases} \mu_{\mathbb{L}}([\boldsymbol{\theta_{t-1}}, \overline{\mathbf{0}}], \boldsymbol{\nu} = [1 - \gamma\lambda, \gamma\lambda]) & \text{if } \boldsymbol{\theta} \in \mathbb{L} \\ \boldsymbol{\theta_{t-1}} - \gamma\boldsymbol{\theta_{t-1}}\lambda & \text{otherwise} \end{cases}$$

where $\mu_{\mathbb{L}}$ is the Lorentz centroid, and $\boldsymbol{\nu}$ are the weights. By removing the later gradient decay and introducing this operation, we adapt AdamW for use in the Lorentz space.

---

**Algorithm 1** Curvature Learning Aware Optimization

---

Given parameters $\boldsymbol{\theta} = [\boldsymbol{\theta}_{euclid}, \boldsymbol{\theta}_{\mathbb{L}}, \boldsymbol{\theta}_K]$ and gradients denoted $\mathcal{G}$
**procedure** OPTIMIZER STEP($\boldsymbol{\theta}$)
    **for** $p$ in $\boldsymbol{\theta}_{\mathbb{L}}$ **do**
        Vanilla Riemannian Optimizer Step
    **end for**
    **for** $p$ in $\boldsymbol{\theta}_{euclid} \cup \boldsymbol{\theta}_K$ **do**
        Vanilla Eucldiean Optimizer Step
    **end for**
    MoveParameters()
**end procedure**
**procedure** MOVE PARAMETERS
    **for** $p$ in $\boldsymbol{\theta}_{\mathbb{L}}$ **do**
        $\mathcal{G}_{temp} = \text{PT}_{\boldsymbol{p} \to \overline{\mathbf{0}}_{t-1}}^{K_{t-1}}(\mathcal{G})$
        $\boldsymbol{z} = \log_{\overline{\mathbf{0}}_{t-1}}^{K_{t-1}}(\boldsymbol{p})$
        $\boldsymbol{p} = \exp_{\overline{\mathbf{0}}_t}^{K_t}(\boldsymbol{z})$
        $\mathcal{G} = \text{PT}_{\overline{\mathbf{0}}_t \to \boldsymbol{p}}^{K_t}(\mathcal{G}_{temp})$
    **end for**
**end procedure**

---

**Maximum Distance Rescaling** Vectors in the hyperboloid models can be defined as $\boldsymbol{x} = [x_t, \boldsymbol{x_s}]^T \in \mathbb{L}_K^n$ where $x_t = \sqrt{||\boldsymbol{x_s}||^2 + K}$, $K = -1/c$ and $c$ is the manifold curvature. As such, Lorentzian projections and operations rely on the ability to accurately calculate the corresponding time component $x_t$ for the hyperbolic vectors. Under Float64 precision, Mishne et al. (2022) derive a maximum value for the time component $x_{t_{max}} = 10^8$. Values above this push vectors off the Lorentz manifold and onto the cone defined by $x_t^2 = \sum \boldsymbol{x_s}^2$. Based on the above, and given a specific $K$, we can derive a maximum representational radius for the model as

$$D_{\overline{\mathbf{0}}_{max}}^K = arccosh(\frac{x_{t_{max}}}{\sqrt{K}}) \cdot \sqrt{K} \qquad (1)$$

Under Float32 precision, and to account for values of $K < 1$ we use a limit value of $x_{t_{max}} = 2 \cdot 10^3$. When projected onto the tangent space of the origin, and with $K = 1$, this translates to $||\log_{\overline{\mathbf{0}}}^1(\boldsymbol{x})|| = D_{\overline{\mathbf{0}}_{max}}^1 = 9.1$. This value changes considerably as the value of $K$ changes. Vectors outside this radius lead to instability and performance degradation due to inaccurate approximation. This problem is only exacerbated as the dimensionality of the hyperbolic vector increases. Higher dimensional vectors tend to have larger norms which limits hyperbolic models' abilities to scale up.

To constrain hyperbolic vectors within a specified maximum distance, either a normalization function or a parameter clipping method is required. Parameter clipping can be challenging to train, as it may lead to information loss and introduce non-smooth gradients. On the other hand, common normalization functions like tanh and the sigmoid function tend to saturate quickly, limiting their effectiveness as seen in the sigmoid implementation by Chen et al. (2022). To address these issues,

we introduce a modified scaling function, designed to provide finer control over both the maximum values and the slope of the curve. A visualization of this function is provided in Figure 3, and the formulation is presented below:

$$\boldsymbol{y}_{rescaled} = \frac{\boldsymbol{y}}{\|\boldsymbol{y}\|} \cdot m \cdot tanh(\|\boldsymbol{y}\| \cdot \frac{atanh(0.99)}{s \cdot m}) \tag{2}$$

where $y \in \mathbb{R}^d$, $m$ is our desired maximum value, and $s$ controls the slope of the curve. We now have a maximum distance value to adhere to and a flexible distance normalizing function. To apply this to the hyperbolic vectors, we suggest performing the scaling on the tangent plane of the origin. However, this is an expensive operation to perform often, as such we derive in appendix A.2 the equivalent factorized form for the scaling of the space values:

$$\boldsymbol{x}_{s_{rescaled}} = \boldsymbol{x}_s \times \frac{e^{\frac{D(\boldsymbol{x},\overline{\boldsymbol{0}})^K_{rescaled}}{\sqrt{K}}} - e^{\frac{-D(\boldsymbol{x},\overline{\boldsymbol{0}})^K_{rescaled}}{\sqrt{K}}}}{e^{\frac{D(\boldsymbol{x},\overline{\boldsymbol{0}})^K}{\sqrt{K}}} - e^{\frac{-D(\boldsymbol{x},\overline{\boldsymbol{0}})^K}{\sqrt{K}}}} \tag{3}$$

where $D(\boldsymbol{x},\overline{\boldsymbol{0}})^k_{rescaled}$ is the scaled distances by plugging in $D(\boldsymbol{x},\overline{\boldsymbol{0}})$ and $D^K_{\overline{\boldsymbol{0}}_{max}}$ in Eq.2. The time component of the vector is then recalculated based on the norm of its spatial components. This gives us a complete scaling operation for the Lorentz space. We apply this tanh scaling when moving parameters across different manifolds. This includes transitions from the Euclidean space to the Lorentz space, as well as between Lorentz spaces of different curvatures. Additionally, we use this scaling after Lorentz Boosts and direct Lorentz concatenations (Qu & Zou, 2022). We also incorporate it following the variance-based rescaling in the batch normalization layer, since variance adjustments during this operation can push points outside the radius during the operation and often lead to invalid values.

### 3.3 Towards Efficient Architectural Components

**Lorentz Convolutional Layer**   In their work, Bdeir et al. (2024) proposed a fully hyperbolic 2D convolutional layer by breaking down the convolution operation into a window-unfolding step followed by a modified version of the Lorentz Linear Layer from Chen et al. (2022). This approach ensured that the convolution outputs remained on the hyperboloid. However, the manual patch creation combined with matrix multiplication made the computation extremely expensive, as it prevented the use of highly optimized CUDA implementations for convolutions.

To address this issue, we adopt an alternative definition of the Lorentz Linear layer from Dai et al. (2021), which decomposes the transformation into a Lorentz boost and a Lorentz rotation. Using this definition, we replace the matrix multiplication employed by Bdeir et al. (2024) for the spatial dimensions and time component projection with a learned rotation operation and a Lorentz boost. Additionally, we can achieve the rotation operation using a parameterization of the convolution weights while still relying on the CUDA convolution implementations, significantly improving computational efficiency.

A rotation is any operation that modifies the directions of transformed vectors while preserving their magnitudes. This can be achieved using a matrix $\hat{\boldsymbol{W}} \in \mathbb{S}$. Where

$$\mathbb{S}(n', n) = \boldsymbol{M} \in \mathbb{R}^{(n' \cdot n)} : \boldsymbol{M}^T \boldsymbol{M} = \boldsymbol{I} \tag{4}$$

represents the Stiefel manifold. The condition $n \leq n'$ ensures that the columns of the weight matrix are orthogonal, thereby preserving the norms along the rows of the input matrix, which correspond to the individual embeddings in this context.

To apply this concept to the convolution operation, the convolution weights, after unfolding, must form a rotation matrix. We define the dimensions of this matrix as $n = (channels_{in} \cdot kernel_{width} \cdot kernel_{height})$ and $n' = channels_{out}$ respectively. However, the condition for orthogonal columns is not always satisfied as there will be instances where $n > n'$. For these cases we use the norm-preserving transformation $\boldsymbol{z} = \boldsymbol{W}^T \boldsymbol{x} \cdot \frac{\|\boldsymbol{x}\|}{\|\boldsymbol{W}^T \boldsymbol{x}\|}$ where $\boldsymbol{W} \in \mathbb{R}^{(n' \cdot n)}$. This formulation allows us to utilize existing efficient implementations of the convolution operation by directly parameterizing

the kernel weights before passing them into the convolutional layer. Finally, we formalize the new Lorentz Convolution as:

$$\boldsymbol{out} = \text{LorentzBoost}(\text{TanhScaling}(\text{RotationConvolution}(\boldsymbol{x}))) \qquad (5)$$

where TanhRescaling is the operation described in Eq.3 and RotationConvolution is a normal convolution parameterized through the procedure in Algorithm 2, where Orthogonalize is a Cayley projection similar to (Helfrich et al., 2018) or the norm-preserving transformation above. We specifically use the Cayley projection because it always produces an orthonormal matrix with a positive determinant, ensuring that the rotated point remains on the upper sheet of the hyperboloid and avoids being mapped to the lower sheet.

$\hat{\boldsymbol{W}}_{core}$ can also be learned directly on the SPD manifold similar to Dai et al. (2021). This definition of the convolution operation allows to use the existing efficient implementations of 2D convolutions, saving both memory and runtime.

**Lorentz-Core Bottleneck Block** To build on the concept of hybrid hyperbolic encoders introduced by (Bdeir et al., 2024), we developed the Lorentz Core Bottleneck blocks for hyperbolic ResNet-based models. These blocks are similar to standard Euclidean bottleneck blocks but replace the internal 3x3 convolutional

---

**Algorithm 2** Lorentz Convolution Parametarization

---

$\boldsymbol{W} \in \mathbb{R}^{C_{in}, C_{out}, K_{width}, K_{length}}$
**procedure** ADAPTWEIGHT(W)
    **if** $K_{width} \cdot K_{length} \cdot C_{in} > C_{out}$ **then**
        return $\boldsymbol{W}$
    **else**
        $\boldsymbol{W} = \boldsymbol{W}.reshape(K_{width} \cdot K_{length} \cdot C_{in}, C_{out})$
        $\hat{\boldsymbol{W}}_{core} = \text{Orthogonalize}(\boldsymbol{W})$
        return $\hat{\boldsymbol{W}}.reshape(C_{in}, C_{out}, K_{width}, K_{length})$
    **end if**
**end procedure**

---

layer with our efficient convolutional layer as illustrated in figure 2. This design allows us to incorporate hyperbolic structuring of the embeddings within each block while retaining the flexibility and computational efficiency of Euclidean models. We interpret this integration as a form of hyperbolic bias, enabling ResNets to leverage hyperbolic representations without requiring fully hyperbolic modeling.

## 4 EXPERIMENTS

### 4.1 HIERARCHICAL METRIC LEARNING PROBLEM

In their paper Kim et al. (2023) take on the problem of hierarchical clustering using an unsupervised hyperbolic loss regularizer they name HIER. This method relies on the use of hierarchical proxies as learnable ancestors of the embedded data points in hyperbolic space. Given a triplet of points $xi, xj, xk$ where $x_i$ and $x_j$ are determined to be related by a reciprocal nearest neighbor measure, and $x_k$ is an unrelated point, the HIER loss regularizer is then calculated as

$$\begin{aligned}
\mathcal{L}_{\text{HIER}}(t) = &[D_B(x_i, \rho_{ij}) - D_B(x_i, \rho_{ijk}) + \delta]_+ \\
&+ [D_B(x_j, \rho_{ij}) - D_B(x_j, \rho_{ijk}) + \delta]_+ \\
&+ [D_B(x_k, \rho_{ijk}) - D_B(x_k, \rho_{ij}) + \delta]_+,
\end{aligned} \qquad (6)$$

where $D_B$ denotes the hyperbolic distance on the Poincaré ball, and $\rho_{ij}$ is the most likely least common ancestor of points $x_i$ and $x_j$. This encourages a smaller hyperbolic distance between $x_i, x_j$, and $\rho_{ij}$, and a larger distance with $\rho_{ijk}$. The opposite signal is then applied in the case of $x_k$, the irrelevant data point. Kim et al. (2023) show substantial performance uplifts for the HIER loss when applied to a variety of network architectures.

Figure 2: Lorentz-Core Bottleneck Block

In the following experiment, we extend HIER to the Lorentz model (LHIER) and compare against the results provided by Kim et al. (2023).

**Experimental Setup** We follow the experimental setup in Kim et al. (2023) and rely on four main datasets: CUB-200-2011 (CUB)(Welinder et al., 2010), Cars-196 (Cars)(Krause et al., 2013), Stanford Online Product (SOP)(Song et al., 2016), and In-shop Clothes Retrieval (InShop)(Liu et al., 2016). Performance is measured using Recall@k which is the fraction of queries with one or more relevant samples in their k-nearest neighbors. Additionally, all model backbones are pre-trained on Imagenet to ensure fair comparisons with the previous work.

**Moving to Lorentz Space** To adapt the HIER model to the hyperboloid we first replace the Euclidean linear layer with a Lorentzian linear layer in the model neck and implement our max distance scaling operation after. We then set the hierarchical proxies as learnable hyperbolic parameters and optimize them directly on the manifold using our Lorentzian AdamW. Finally, we change the Poincaré distance to the Lorentz distance for the LHIER loss and set the hierarchical proxies to be scaled beforehand. We continue to use FP16 precision during curvature learning to evaluate the stability of the new optimization scheme. Our experiments also include both CNN-based and transformer models to assess whether differences in architecture affect the performance of the hyperbolic heads.

**Results** As shown in table 1, our HIER+ model manages to improve the performance of mainly the CNN-based models with improvements ranging in recall@1 from 1-2.3%. In contrast, transformer models exhibit marginal gains in most cases and even show slight declines in other scenarios. This indicates that the proposed components do not better fully exploit the advantages of transformer architectures compared to other hyperbolic methods. The reasons may include architectural limitations or insufficient hyperparameter tuning. Notably, Kim et al. (2023) employed an extensive set of hyperparameters for model training, which we kept at default values to ensure fair comparisons and avoid costly hyperparameter searches.

## 4.2 STANDARD CLASSIFICATION PROBLEM

**Experimental Setup** We follow the experimental setup and hyperparameters of Bdeir et al. (2024) and rely on three main datasets: CIFAR10, CIFAR100, and Mini-Imagenet. We denote HCNN+ as the fully hyperbolic model from Bdeir et al. (2024), updated with efficient convolution implementations and our proposed scaling function. This experiment aims to boost model efficiency and address the performance inconsistencies observed in the original version.

For ResNet-18, we employ HECNN+, following the structure outlined in Bdeir et al. (2024) with alternating Euclidean and hyperbolic blocks, and integrate our efficient convolution operation. While we do not expect similar efficiency gains to those seen in fully hyperbolic models, our focus here is on enhancing scalability. In the case of ResNet-50, we adapt HECNN+ by replacing all blocks with the Lorentz-Core bottleneck block, where we anticipate the most significant efficiency improvements due to the larger model size. Our goal is to evaluate if the Lorentz-Core block can maintain strong performance under these conditions.

For both ResNet-18 and ResNet-50, we use Riemannian SGD with our improved learning scheme and curvature learning. We also decouple the encoder and decoder manifolds, allowing each to independently learn its own curvature for enhanced model flexibility.

**Results** For the ResNet-18 experiments, Table 2 shows that the new architectures achieve comparable or better results across almost all cases. The smallest improvements are observed with the hybrid models, likely due to the reduced impact of the hyperbolic components compared to a fully hyperbolic model. However, we see a significant performance gap between the fully hyperbolic models, where our proposed architecture now matches the performance of the hybrid encoders. We hypothesize that the improved scaling helps mitigate the previous performance inconsistencies, with the whole model being two times more memory efficient.

In the ResNet-50 experiments in Table 3, we observe that HECNN+ significantly outperforms the Euclidean model across all datasets, including Tiny-ImageNet, where other models typically begin to show a drop in accuracy. This improvement is likely due to the tighter integration of hyperbolic components and the enhanced scaling, which helps manage the challenges of higher-dimensional embeddings.

We evaluate the impact of our efficient convolution and Lorentz-Core block in Table 3. We see a $\sim 48\%$ reduction in memory usage and $\sim 66\%$ reduction in runtime. We attribute this improvement

Table 1: Performance of metric learning methods on the four datasets as provided by (Kim et al., 2023). All architecture backbones are pretrained and tested with the new LHIER loss. Superscripts denote their embedding dimensions and † indicates models using larger input images. As in (Kim et al., 2023), network architectures are abbreviated as, R–ResNet50 (He et al., 2016), B–Inception with BatchNorm (Ioffe & Szegedy, 2015), De–DeiT (Touvron et al., 2021), DN–DINO (Caron et al., 2021) and V–ViT (Dosovitskiy et al., 2021)

| Methods | Arch. | CUB | | | Cars | | | SOP | | |
|---|---|---|---|---|---|---|---|---|---|---|
| | | R@1 | R@2 | R@4 | R@1 | R@2 | R@4 | R@1 | R@10 | R@100 |
| *Backbone architecture*: *CNN* | | | | | | | | | | |
| NSoftmax (Zhai & Wu, 2018) | $R^{128}$ | 56.5 | 69.6 | 79.9 | 81.6 | 88.7 | 93.4 | 75.2 | 88.7 | 95.2 |
| MIC (Roth et al., 2019) | $R^{128}$ | 66.1 | 76.8 | 85.6 | 82.6 | 89.1 | 93.2 | 77.2 | 89.4 | 94.6 |
| XBM (Wang et al., 2020) | $R^{128}$ | - | - | - | - | - | - | 80.6 | 91.6 | 96.2 |
| XBM (Wang et al., 2020) | $B^{512}$ | 65.8 | 75.9 | 84.0 | 82.0 | 88.7 | 93.1 | 79.5 | 90.8 | 96.1 |
| HTL (Ge et al., 2018) | $B^{512}$ | 57.1 | 68.8 | 78.7 | 81.4 | 88.0 | 92.7 | 74.8 | 88.3 | 94.8 |
| MS (Wang et al., 2019) | $B^{512}$ | 65.7 | 77.0 | 86.3 | 84.1 | 90.4 | 94.0 | 78.2 | 90.5 | 96.0 |
| SoftTriple (Qian et al., 2019) | $B^{512}$ | 65.4 | 76.4 | 84.5 | 84.5 | 90.7 | 94.5 | 78.6 | 86.6 | 91.8 |
| PA (Kim et al., 2020) | $B^{512}$ | 68.4 | 79.2 | 86.8 | 86.1 | 91.7 | 95.0 | 79.1 | 90.8 | 96.2 |
| NSoftmax (Zhai & Wu, 2018) | $R^{512}$ | 61.3 | 73.9 | 83.5 | 84.2 | 90.4 | 94.4 | 78.2 | 90.6 | 96.2 |
| †ProxyNCA++ (Teh et al., 2020) | $R^{512}$ | 69.0 | 79.8 | 87.3 | 86.5 | 92.5 | 95.7 | 80.7 | 92.0 | 96.7 |
| Hyp (Ermolov et al., 2022) | $R^{512}$ | 65.5 | 76.2 | 84.9 | 81.9 | 88.8 | 93.1 | 79.9 | 91.5 | 96.5 |
| HIER (Kim et al., 2023) | $R^{512}$ | 70.1 | 79.4 | 86.9 | 88.2 | 93.0 | 95.6 | 80.2 | 91.5 | 96.6 |
| LHIER (ours) | $R^{512}$ | **72.4** | **81.5** | **88.4** | **89.1** | **93.5** | **96.1** | **81.3** | **92.1** | **96.8** |
| *Backbone architecture*: *ViT* | | | | | | | | | | |
| IRT$_R$ (El-Nouby et al., 2021) | $De^{128}$ | 72.6 | 81.9 | 88.7 | - | - | - | **83.4** | 93.0 | 97.0 |
| Hyp (Ermolov et al., 2022) | $De^{128}$ | 74.7 | 84.5 | 90.1 | 82.1 | 89.1 | 93.4 | 83.0 | 93.4 | **97.5** |
| HIER (Kim et al., 2023) | $De^{128}$ | 75.2 | 84.2 | 90.0 | 85.1 | 91.1 | 95.1 | 82.5 | 92.7 | 97.0 |
| LHIER (ours) | $De^{128}$ | **75.5** | **84.7** | **90.6** | **85.4** | **91.7** | **95.6** | 82.7 | **93.5** | 97.4 |
| Hyp (Ermolov et al., 2022) | $DN^{128}$ | 78.3 | 86.0 | 91.2 | 86.0 | 91.9 | 95.2 | 84.6 | 94.1 | 97.7 |
| HIER (Kim et al., 2023) | $DN^{128}$ | 78.5 | 86.7 | 91.5 | 88.4 | **93.3** | 95.9 | 84.9 | 94.2 | 97.5 |
| LHIER (ours) | $DN^{128}$ | **78.8** | **87.0** | **91.9** | **88.9** | 93.2 | **96.4** | **85.1** | **94.9** | **98.2** |
| Hyp (Ermolov et al., 2022) | $V^{128}$ | 84.0 | 90.2 | 94.2 | 82.7 | 89.7 | 93.9 | 85.5 | 94.9 | **98.1** |
| HIER (Kim et al., 2023) | $V^{128}$ | 84.2 | 90.1 | 93.7 | 86.4 | 91.9 | 95.1 | 85.6 | 94.6 | 97.8 |
| LHIER (ours) | $V^{128}$ | **84.6** | **90.2** | **93.9** | **86.7** | **92.0** | **95.3** | **85.9** | **95.0** | 98.0 |
| IRT$_R$ (El-Nouby et al., 2021) | $De^{384}$ | 76.6 | 85.0 | 91.1 | - | - | - | 84.2 | 93.7 | 97.3 |
| DeiT-S (Touvron et al., 2021) | $De^{384}$ | 70.6 | 81.3 | 88.7 | 52.8 | 65.1 | 76.2 | 58.3 | 73.9 | 85.9 |
| Hyp (Ermolov et al., 2022) | $De^{384}$ | 77.8 | 86.6 | 91.9 | 86.4 | 92.2 | 95.5 | **83.3** | **93.5** | **97.4** |
| HIER (Kim et al., 2023) | $De^{384}$ | **78.7** | **86.8** | **92.0** | **88.9** | **93.9** | **96.6** | 83.0 | 93.1 | 97.2 |
| LHIER (ours) | $De^{384}$ | 78.3 | 86.2 | 91.8 | 88.7 | 93.4 | 96.4 | 82.8 | 92.9 | 96.9 |
| DINO (Caron et al., 2021) | $DN^{384}$ | 70.8 | 81.1 | 88.8 | 42.9 | 53.9 | 64.2 | 63.4 | 78.1 | 88.3 |
| Hyp (Ermolov et al., 2022) | $DN^{384}$ | 80.9 | 87.6 | 92.4 | 89.2 | 94.1 | 96.7 | 85.1 | 94.4 | 97.8 |
| HIER (Kim et al., 2023) | $DN^{384}$ | 81.1 | 88.2 | 93.3 | 91.3 | **95.2** | 97.1 | 85.7 | 94.6 | 97.8 |
| LHIER (ours) | $DN^{384}$ | **81.3** | **88.4** | **93.3** | **91.5** | 95.1 | **97.6** | **85.9** | **95.0** | **98.1** |
| ViT-S (Dosovitskiy et al., 2021) | $V^{384}$ | 83.1 | 90.4 | 94.4 | 47.8 | 60.2 | 72.2 | 62.1 | 77.7 | 89.0 |
| Hyp (Ermolov et al., 2022) | $V^{384}$ | 85.6 | 91.4 | 94.8 | 86.5 | 92.1 | 95.3 | 85.9 | 94.9 | 98.1 |
| HIER (Kim et al., 2023) | $V^{384}$ | 85.7 | 91.3 | 94.4 | 88.3 | 93.2 | 96.1 | 86.1 | 95.0 | 98.0 |
| LHIER (ours) | $V^{384}$ | **86.2** | **92.1** | **95.2** | **88.6** | **93.6** | **96.3** | **86.7** | **95.3** | **98.3** |

to the efficient closed-source convolution operations we can now leverage. Similar gains are also observed in generation tasks and classification with smaller models. However, there remains significant room for further optimization when compared to the Euclidean baseline. A key remaining performance bottleneck identified is the batch normalization step, which contributes approximately 60% of the runtime and around 30% of the memory usage. The next step would be to factorize the extensive parallel transports and tangent mappings involved in this operation, potentially alleviating the associated overhead.

## 4.3 VAE IMAGE GENERATION

**Experimental Setup** Previous research has highlighted the ability of hyperbolic neural networks to perform better when using lower dimensional embeddings due to the more expressive space (Nagano

Table 2: Classification accuracy (%) of ResNet-18 models. We estimate the mean and standard deviation from five runs. The best performance is highlighted in bold (higher is better).

| | CIFAR-10 $(\delta_{rel} = 0.26)$ | CIFAR-100 $(\delta_{rel} = 0.23)$ | VRAM/$t_{epoch}$ For Cifar100 | Tiny-ImageNet $(\delta_{rel} = 0.20)$ |
|---|---|---|---|---|
| Euclidean (He et al., 2016) | $95.14_{\pm 0.12}$ | $77.72_{\pm 0.15}$ | 1.2GB - 12s | $65.19_{\pm 0.12}$ |
| Hybrid Poincaré (Guo et al., 2022) | $95.04_{\pm 0.13}$ | $77.19_{\pm 0.50}$ | - | $64.93_{\pm 0.38}$ |
| Hybrid Lorentz (Bdeir et al., 2024) | $94.98_{\pm 0.12}$ | $78.03_{\pm 0.21}$ | - | $65.63_{\pm 0.10}$ |
| Poincaré ResNet (van Spengler et al., 2023) | $94.51_{\pm 0.15}$ | $76.60_{\pm 0.32}$ | - | $62.01_{\pm 0.56}$ |
| HECNN Lorentz (Bdeir et al., 2024) | $95.16_{\pm 0.11}$ | $78.76_{\pm 0.24}$ | 4.3GB - 100s | $65.96_{\pm 0.18}$ |
| HECNN+ (ours) | $95.15_{\pm 0.07}$ | $78.80_{\pm 0.12}$ | 3GB - 80s | $65.98_{\pm 0.11}$ |
| HCNN Lorentz (Bdeir et al., 2024) | $95.15_{\pm 0.08}$ | $78.07_{\pm 0.17}$ | 10GB - 175s | $65.71_{\pm 0.13}$ |
| HCNN+ (ours) | $95.17_{\pm 0.09}$ | $\mathbf{78.81_{\pm 0.19}}$ | 5GB - 140s | $\mathbf{66.12_{\pm 0.14}}$ |

Table 3: Classification accuracy (%) of ResNet-50 models. The best performance is highlighted in bold (higher is better).

| | CIFAR-10 $(\delta_{rel} = 0.26)$ | CIFAR-100 $(\delta_{rel} = 0.23)$ | VRAM/$t_{epoch}$ | Tiny-ImageNet $(\delta_{rel} = 0.20)$ |
|---|---|---|---|---|
| Euclidean (He et al., 2016) | 95.14 | 78.52 | 4.5GB - 30s | 66.23 |
| Hybrid Lorentz (Bdeir et al., 2024) | 95.38 | 79.35 | - | 66.01 |
| HECNN (Bdeir et al., 2024) | 95.42 | 79.83 | 15.6GB - 300s | 66.30 |
| HCNN+ w Bottleneck Conv. (ours) | **95.46** | **80.86** | 8.1GB - 100s | **67.18** |

et al., 2019; Mathieu et al., 2019; Ovinnikov, 2019; Hsu et al., 2020). A natural extension to this would be implementing the models for VAEs which rely on smaller latent embedding dimensions to encode the inputs. Bdeir et al. (2024) perform this study on image generation and reconstruction for their fully hyperbolic models. In the following section, we reproduce the experimental setup and re-implement the fully hyperbolic VAE using our new efficient convolution and transpose convolution layers. We also use curvature learning with our adjusted Riemannian SGD learning scheme.

**Results** We see in table 4 that our implementation of the fully hyperbolic VAE achieves better performance on both datasets. It should also be noted that this is achieved with around 2.5x less memory and at 3x greater training speed. This helps demonstrate both the effectiveness of our proposed curvature learning process and our efficient model components.

Table 4: Reconstruction and generation FID of manifold VAEs across five runs (lower is better).

| | CIFAR-100 | | CelebA | |
|---|---|---|---|---|
| | Rec. FID | Gen. FID | Rec. FID | Gen. FID |
| Euclidean | $63.81_{\pm 0.47}$ | $103.54_{\pm 0.84}$ | $54.80_{\pm 0.29}$ | $79.25_{\pm 0.89}$ |
| Hybrid Poincaré (Mathieu et al., 2019) | $62.64_{\pm 0.43}$ | $98.19_{\pm 0.57}$ | $54.62_{\pm 0.61}$ | $81.30_{\pm 0.56}$ |
| Hybrid Lorentz (Nagano et al., 2019) | $62.14_{\pm 0.35}$ | $98.34_{\pm 0.62}$ | $54.64_{\pm 0.34}$ | $82.78_{\pm 0.93}$ |
| HCNN Lorentz (Bdeir et al., 2024) | $61.44_{\pm 0.64}$ | $100.27_{\pm 0.84}$ | $54.17_{\pm 0.66}$ | $78.11_{\pm 0.95}$ |
| HCNN+ Lorentz (Ours) | $\mathbf{57.69_{\pm 0.52}}$ | $\mathbf{98.14_{\pm 0.44}}$ | $\mathbf{52.73_{\pm 0.27}}$ | $\mathbf{77.98_{\pm 0.32}}$ |

## 5 CONCLUSION

In our work, we present many new components and schemas for the use of hyperbolic deep learning in hyperbolic vision. We test these components in three vision tasks and prove the potential of these new components even in float16 conditions.

However, there is still significant room for improvement. Further optimizations to the batch normalization layers could enhance the efficiency of hyperbolic models. Additionally, a key challenge remains with the hyperbolic linear layers when reducing dimensionality. Currently, we match norms to simulate a rotation operation, but we encourage exploring alternative approaches that align more naturally with the mathematical properties of the manifold.

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

## A  APPENDIX

### A.1  OPERATIONS IN HYPERBOLIC GEOMETRY

**Parallel Transport**    A parallel transport operation $\text{PT}_{\boldsymbol{x} \to \boldsymbol{y}}^{K}(\boldsymbol{v})$ describes the mapping of a vector on the manifold $\boldsymbol{v}$ from the tangent space of $\boldsymbol{x} \in \mathbb{L}$ to the tangent space of $\boldsymbol{y} \in \mathbb{L}$. This operation is given as $\text{PT}_{\boldsymbol{x} \to \boldsymbol{y}}^{K}(\boldsymbol{v}) = \boldsymbol{v} + \frac{\langle \boldsymbol{y}, \boldsymbol{v} \rangle_{\mathbb{L}}}{K - \langle \boldsymbol{x}, \boldsymbol{y} \rangle_{\mathbb{L}}}(\boldsymbol{x} + \boldsymbol{y})$.

**Lorentzian Centroid (Law et al., 2019)**    Also denoted as $\boldsymbol{\mu}_{\mathbb{L}}$, is the weighted centroid between points on the manifold based on the Lorentzian square distance. Given the weights $\boldsymbol{\nu}$, $\boldsymbol{\mu} = \frac{\sum_{i=1}^{m} \nu_i \boldsymbol{x}_i}{\sqrt{1/K} \left| \| \sum_{i=1}^{m} \nu_i \boldsymbol{x}_i \|_{\mathcal{L}} \right|}$.

**Lorentz Transformations**  In the Lorentz model, linear transformations preserving the structure of spacetime are termed Lorentz transformations. A matrix $\mathbf{A}^{(n+1)\times(n+1)}$ is defined as a Lorentz transformation if it provides a linear mapping from $\mathbb{R}^{n+1}$ to $\mathbb{R}^{n+1}$ that preserves the inner product, i.e., $\langle \mathbf{A}\boldsymbol{x}, \mathbf{A}\boldsymbol{y}\rangle_{\mathbb{L}} = \langle \boldsymbol{x}, \boldsymbol{y}\rangle_{\mathbb{L}}$ for all $\boldsymbol{x}, \boldsymbol{y} \in \mathbb{R}^{n+1}$. The collection of these matrices forms an orthogonal group, denoted $\boldsymbol{O}(1, n)$, which is commonly referred to as the Lorentz group.

In this model, we restrict attention to transformations that preserve the positive time orientation, operating within the upper sheet of the two-sheeted hyperboloid. Accordingly, the transformations we consider lie within the positive Lorentz group, denoted $\boldsymbol{O}^+(1, n) = \mathbf{A} \in \boldsymbol{O}(1, n) : a_{11} > 0$, ensuring preservation of the time component sign $x_t$ for any $\boldsymbol{x} \in \mathbb{L}_K^n$. Specifically, in this context, Lorentz transformations satisfy the relation

$$\boldsymbol{O}^+(1, n) = \mathbf{A} \in \mathbb{R}^{(n+1)\times(n+1)} | \forall \boldsymbol{x} \in \mathbb{L}_K^n : \langle \mathbf{A}\boldsymbol{x}, \mathbf{A}\boldsymbol{x}\rangle_{\mathbb{L}} = -\frac{1}{K}, (\mathbf{A}\boldsymbol{x})_0 > 0). \tag{7}$$

Each Lorentz transformation can be decomposed via polar decomposition into a Lorentz rotation and a Lorentz boost, expressed as $\mathbf{A} = \mathbf{R}\mathbf{B}$ Moretti (2002). The rotation matrix $\mathbf{R}$ is designed to rotate points around the time axis and is defined as

$$\mathbf{R} = \begin{bmatrix} 1 & \mathbf{0}^T \\ \mathbf{0} & \tilde{\mathbf{R}} \end{bmatrix}, \tag{8}$$

where $\mathbf{0}$ represents a zero vector, $\tilde{\mathbf{R}}$ satisfies $\tilde{\mathbf{R}}^T\tilde{\mathbf{R}} = \mathbf{I}$, and $\det(\tilde{\mathbf{R}}) = 1$. This structure shows that Lorentz rotations on the upper hyperboloid sheet belong to a special orthogonal subgroup, $\boldsymbol{SO}^+(1, n)$, which preserves orientation, with $\tilde{\mathbf{R}} \in \boldsymbol{SO}(n)$.

In contrast, the Lorentz boost applies shifts along spatial axes given a velocity vector $\boldsymbol{v} \in \mathbb{R}^n$ with $||\boldsymbol{v}|| < 1$, without altering the time axis.

$$\mathbf{B} = \begin{bmatrix} \gamma & -\gamma\boldsymbol{v}^T \\ -\gamma\boldsymbol{v} & \mathbf{I} + \frac{\gamma^2}{1+\gamma}\boldsymbol{v}\boldsymbol{v}^T \end{bmatrix}, \tag{9}$$

with $\gamma = \frac{1}{\sqrt{1-||\boldsymbol{v}||^2}}$. However, this can also be any operation that scales the norms of the space values without changing the vector orientation.

## A.2  Scaling Lorentzian Vectors

**Tanh Scaling**  We show the output of the tanh scaling function in Figure 3. By changing the transformation parameters we are able to fine-tune the maximum output and the slope to match our desired function response.

**Hyperbolic Scaling**  We isolate the transformation of the $\exp_{\mathbf{0}}^K(y)$ operation on the space values of $y$ as:

$$\boldsymbol{x}_s = \sqrt{K} \times \sinh(\frac{\|\boldsymbol{y}\|_{\mathbb{L}}}{\sqrt{K}})\frac{\boldsymbol{y}}{\|\boldsymbol{y}\|_{\mathbb{L}}} \tag{10}$$

where $\boldsymbol{y} \in \mathbb{R}^d = \log_{\mathbf{0}}^K(\boldsymbol{x})$. However, at the tangent plane of the origin, the first element $\boldsymbol{y}_0$ becomes 0. As such $\|\boldsymbol{y}\|_{\mathbb{L}} = \|\boldsymbol{y}\|_{\mathbb{E}} = \sum_{i=2}^d \boldsymbol{y_i^2}$. This gives us:

$$\boldsymbol{x}_s = \sqrt{K} \times \sinh(\frac{\|\boldsymbol{y}\|_{\mathbb{E}}}{\sqrt{K}})\frac{\boldsymbol{y}}{\|\boldsymbol{y}\|_{\mathbb{E}}} \tag{11}$$

We can now scale the norm of the Euclidean vector $\boldsymbol{y}$ bay a value $a$ and find the equivalent value for the hyperbolic space elements:

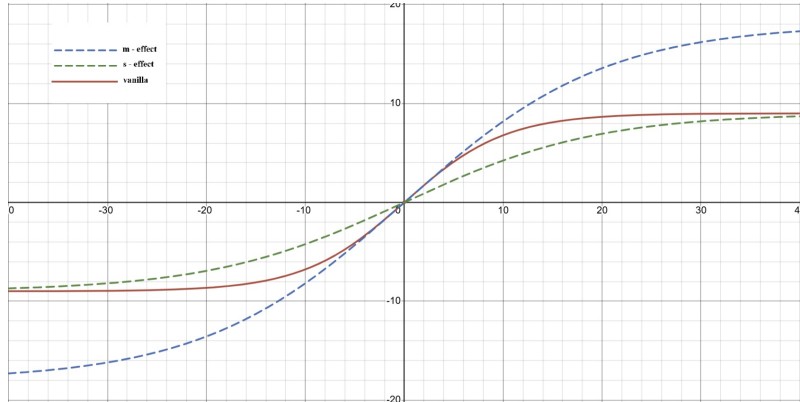

Figure 3: The output of the proposed flexible tanh function. Here the maximum value m is set to 9.1 in the vanilla version with an alternate value of m=18 and the slope s is set to 2.6 with an alternate value of 3.5

$$a_{\mathbb{L}} = \frac{\boldsymbol{x}_{s_{rescaled}}}{\boldsymbol{x}_s} = \frac{sinh(\frac{a \times \|\boldsymbol{y}\|_{\mathbb{E}}}{\sqrt{K}})}{sinh(\frac{\|\boldsymbol{y}\|_{\mathbb{E}}}{\sqrt{K}})} = \frac{e^{\frac{a \times \|\boldsymbol{y}\|_{\mathbb{E}}}{\sqrt{K}}} - e^{\frac{-a \times \|\boldsymbol{y}\|_{\mathbb{E}}}{\sqrt{K}}}}{e^{\frac{\|\boldsymbol{y}\|_{\mathbb{E}}}{\sqrt{K}}} - e^{\frac{-\|\boldsymbol{y}\|_{\mathbb{E}}}{\sqrt{K}}}} \tag{12}$$

Additionally, we know that the hyperbolic distance from the origin of the manifold to any point is equal to the norm of the projected vector onto the tangent plane. Supposing that we want $a \times D(\boldsymbol{x}, \overline{\boldsymbol{0}})^K = D(\boldsymbol{x}, \overline{\boldsymbol{0}})^K_{rescaled}$, we get the final equation:

$$\boldsymbol{x}_{s_{rescaled}} = \boldsymbol{x}_s \times \frac{e^{\frac{D(\boldsymbol{x}, \overline{\boldsymbol{0}})^K_{rescaled}}{\sqrt{K}}} - e^{\frac{-D(\boldsymbol{x}, \overline{\boldsymbol{0}})^K_{rescaled}}{\sqrt{K}}}}{e^{\frac{D(\boldsymbol{x}, \overline{\boldsymbol{0}})^K}{\sqrt{K}}} - e^{\frac{-D(\boldsymbol{x}, \overline{\boldsymbol{0}})^K}{\sqrt{K}}}} \tag{13}$$

## A.3 ABLATIONS

We test the effect of individual model components in table 5. Each subsequent model involves the default architecture presented in the experimental setup minus the mentioned component. As we can see, the best results are achieved when all the architectural components are included. In the case of attempting to learn the curvature without the proposed optimizer schema, the model breaks completely down due to excessive numerical inaccuracies. One other benefit that we find from learning the curvature is quicker convergence. The model is able to reach convergence in 130 epochs vs the 200 epochs required by a static curve model.

Table 5: Ablation experiments for Resnet-50 models on Cifar-100.

|  | CIFAR-100 |
| --- | --- |
| HCNN+ - Default | **80.86** |
| HCNN+ - fixed curve | 79.6 |
| HCNN+ - no scaling | 80.13 |
| HCNN+ - no optim scheme | $NaaN$ |

