# OpenReview forum: "Optimizing Learning for Robust Hyperbolic Deep Learning in Computer Vision"
_ICLR.cc/2025/Conference — Submitted to ICLR 2025_

### Official Review · Reviewer_WUoJ · 2024-10-15

**Soundness:** 3
**Presentation:** 3
**Contribution:** 2
**Rating:** 5
**Confidence:** 4

**Summary:**

This paper presents a set of methods aimed at enhancing hyperbolic deep learning within the context of computer vision. The authors propose improvements to the current hyperbolic learning paradigms, such as introducing a novel Riemannian AdamW optimizer, a bounding method for stable hyperbolic optimization, and an efficient convolutional layer that minimizes computational overhead. The paper demonstrates empirical improvements on tasks including hierarchical metric learning and image classification and demonstrates the improved results after applying their improvements.

**Strengths:**

The introduced components to hyperbolic deep learning are indeed novel, and some of the contributions have some clever ways to work around difficulties in the Riemannian setting, such as the centroid trick for AdamW. The empirical evaluation is extensive and compares against a good number of baselines. It is also nice, especially in the Riemannian optimization community, to see that the authors also consider computational challenges and propose solutions that are efficient and work well with modern hardware.

**Weaknesses:**

(roughly ordered by importance)

- **Insufficient analysis of AdamW in Riemannian context.** The paper draws analogies to AdamW in the Riemannian optimization context, and while the adaptation is interesting, it is fundamentally different than a direct Riemannian AdamW. The adaptation involves changes that are not fully explored or justified, and it remains unclear how closely the benefits of AdamW translate to hyperbolic settings. This seems like an interesting optimization algorithm, but the community would benefit from a more in-depth analysis of these modifications, particularly in understanding why this adaptation should be regarded as a viable alternative to Riemannian Adam.

- **Marginal experimental improvements.** The reported improvements in performance across different tasks are generally quite marginal. In most settings, the proposed approaches show only slight gains over existing methods, which raises questions regarding the significance of these contributions. Since the work is largely empirical, and thus its justification purely comes from demonstrated results, more convincing baselines are needed to demonstrate the effectiveness of these improvements. The self-ablations also show very marginal improvements with each other, not giving insight into the individual role of each component.

- **Lack of theoretical insights.** The paper largely presents itself as an empirical collection of modifications to existing hyperbolic optimization methods. However, there is a lack of rigorous theoretical backing for these modifications. Again, these insights would not be needed if the experiments improved enough to demonstrate their value, but without those it is hard to see the justification to develop and use these improvements, and how future researchers should build off of these.

**Questions:**

1. Are there any theorems about any of the proposed modifications that the authors could provide and give a proof sketch for, that highlight the true value of the improvements?

2. How does the proposed maximum distance rescaling function compare against simpler solutions, such as gradient clipping or norm bounding, in terms of both stability and performance? What drove you all to your particular design choices?

3. Can the authors discuss the practical scenarios where these improvements might be crucial (rather than incremental changes), and how these methods contribute significantly beyond existing alternatives?

---

> ### Author Response · Authors · 2024-11-19
>
> Thank you very much for your feedback on our paper.
>
> __On the Weaknesses:__
>
> 1.	While this was not clear before, we have updated our methodology section to better show that the derived AdamW is a very close adaptation of the Euclidean implementation. We also compared the results to that of Radam and RSGD for HIER and the show the results below:
>
> |        |         | CUB  |         |         | Cars |         |         | SOP  |        |
> | ------ | ---- | ---- | ---- |---- |---- |---- |---- |---- |---- |
> |        | R@1  | R@2  | R@3  | R@1 | R@2 | R@3 | R@1 | R@2 | R@3 |
> | RSGD   | 64.2 | 72.1 | 78.6 | 73.9 | 81.4 | 84.8 | 67.8 | 73.9 | 80.1 |
> | RAdam  | 70.1 | 79.8 | 86.9 | 87.6 | 92.0 | 94.9 | 79.8 | 90.3 | 95.2 |
> | RAdamW | 72.4 | 81.5 | 88.4 | 89.1 | 93.5 | 96.1 | 81.3 | 92.1 | 96.8 |
>
> 2.	Our main aim in this paper is to increase stability and improve the efficiency of the models while maintaining or increasing accuracy. We have updated the results tables to highlight the benefits of our approaches and components, where we get upto 2.5x memory and 3x runtime savings despite performing better or marginally similar. We also manage to learn the curvature in all settings, even float16 precision, without runtime errors, which demonstrates our stability benefits.
> 3.	With the new draft of the paper we have hopefully better explained the theoretical reasoning behind the components in the methodology section and added visualizations for better clarity on key concepts.
>
> __On the Questions:__
>
> 1.	We have added theorems and some mathematical proofs both in the paper and the appendix. Specifically, we have done this for maximum distance rescaling derivation, the mapping, and parallel transport operations used in the new optimizer, and better clarity for the RAdamW and efficient convolution setup.
> 2.	Clipping parameters normally leads to information loss and non-smooth gradients which causes some training issues. Additionally, Chen et al used the sigmoid function to limit the outputs to this radius but Bdeir et al. found that it saturates too quickly which also caused performance issues. Our approach is similar to Chen et al. in that we use a normalization function, we just provide finer control over the saturation limits and mapping slope. Gradient clipping is still used in this paper similar to Bdeir et al.
> 3.	These improvements are crucial for future adaptation into bigger models and datasets. We are now able to run Resnet-50 with reasonable computation requirements for example. Additionally, with a lot of work moving towards float16 and float8 precision, the scaling function becomes crucial since the introduced errors become larger and cause much more instability. This is demonstrated in our work by being able to learn the curvature for the HIER experiments despite it being a float16 environment.

---

> > ### Comment · Reviewer_WUoJ · 2024-11-26
> >
> > Thank you very much for the author's efforts in the rebuttal. I have read through the revised manuscript, but please in the future highlight the changes made in a different color for ease of reading.
> >
> > I appreciate the direct explanation of the AdamW --> RAdamW conversion, and agree the manuscript overall reads better, and I now understand the goal of the paper more. However, while I find the ideas and methods in the paper interesting, I believe their potency has not been properly demonstrated here -- accuracy improvements are marginal, and the ablations+theory section feels still incomplete. The presented ideas/methods are interesting, but as the true roles and pros/cons of the individual components presented in this paper remain unclear for future readers/practitioners, I will keep my score.

---

> > > ### Author Response · Authors · 2024-12-03
> > >
> > > Thank you for going through the rebuttal and the paper once more. We're glad the overall flow of the paper is smoother now and appreciate the interest in the presented methodologies. We agree that maybe too many components were presented which put a limitation on the amount of analysis/experiments we could do/present in the actual limit, but do understand that deeper outlooks would help prove these approaches better.

---

### Official Review · Reviewer_KHqp · 2024-10-22

**Soundness:** 2
**Presentation:** 2
**Contribution:** 3
**Rating:** 6
**Confidence:** 3

**Summary:**

This paper introduces a new method for properly learning the curvature in a hyperbolic learning setting. Previous approaches simply update the curvature of the manifold(s) containing parameters without changing the parameters. This paper rescales the parameters according to the updated curvature of the manifold. Besides this contribution, the paper introduces a hyperbolic version of the AdamW optimizer, a maximum distance rescaling function and a more efficient convolutional layer. The effect of these new proposals is shown through a number of experiments on hierarchical metric learning, image classification, VAE image generation and a number of ablations.

**Strengths:**

The paper addresses an important issue, namely the fact that all previous hyperbolic learning methods that use a learnable curvature ignore the parameters when performing the curvature update, potentially leading to numerical instability. The proposed approach for dealing with this issue seems straightforward and effective. The other contributions are also quite interesting and together form a nice contribution to hyperbolic learning.

The experiments show that the proposed methods indeed lead to a fairly significant improvement compared to existing methods.

**Weaknesses:**

While the ideas behind the paper are great, the presentation of some of these is currently rather unclear. There are also many notational inconsistencies, typos and missing details, which together make the paper (especially the method) difficult to follow and probably impossible to reproduce. Some of the contributions are also missing sufficient motivation in my opinion. Therefore, I think the paper needs some significant revisions before it can be accepted. I will list some of my main concerns here:

Optimizers for learned curvatures:
1. Algorithm 1 is a bit cluttered, making it hard to read. It might be better to just mention the expmap, logmap and PT without giving the full formulations. The formulations of these mappings are already given in the background, so they can easily be substituted if needed.

Maximum distance scaling:
1. For $||x_s|| \gg K$, $x_t \approx ||x_s||$ and the approximation error for float64 is less than $5 * 10^{-9}$. Why can we not simply ignore this rounding error? Why does it lead to problems?
2. Commonly used libraries such as geoopt already perform projection onto the Poincaré ball and hyperboloid after many operations. Why is this projection not enough for numerical stability?
3. I understand that the operation is mapping the input to the tangent space, scaling and mapping back onto the manifold. However, I do not precisely understand where the scaling term is coming from. What exactly is it supposed to achieve? The motivation appears to be coming from the limit that is given after equation 3. However, it is unclear to me how this limit is solved. I think that a visualization of the operations in terms of $||\mathbf{x}||$ versus $||\mathbf{x}_{rescaled}||$ could be a useful aid to help explain this.
4. There are some notational inconsistencies and typos in this section:
    - $D(x, \mathbf{\overline{0}})_{max}^K$ is introduced which hints at a dependency on $x$, but it is not clear what $x$ is in this context. Is this term not a constant? If it is, would it not be better to simply write $D\_{max}^K$?
    - There appear to be some $k$'s that should be $K$'s.
    - Later on a $D\_{0_{max}}^K$ appears, which I am guessing is denoting the same as the earlier $D(x, \mathbf{\overline{0}})_{max}^K$.
    - I guess it should be $||z\_{rescaled}||$ instead of $||z||\_{rescaled}$.
    - Some more small things.

Lorentz convolutional layer:
1. This section introduces a lot of technical stuff without really discussing or motivating it.
2. Lorentz transformations are not discussed in the background, but this section assumes the reader to be familiar with the details of both Lorentz boosts and rotations. The background should include a discussion on these transformations as well, since anyone not familiar with the Lorentz model will almost certainly not be familiar with Lorentz transformations.
3. A stiefel manifold requires $n' \leq n$ (with the notation of the paper). It is an empty set when this is not the case, so maybe it is best to focus on what the goal of the mapping is and not necessarily go into Stiefel manifolds.
4. It is unclear to me what the exact contributions are in this section and which parts are taken from other papers. I am also unsure about the advantages of this convolutional layer with respect to other convolutional layers. Were they originally not compatible with existing 2D convolution implementations? If so, why?
5. Some Caley transform is mentioned with a reference to a paper. As far as I know, within the context of matrices, a Cayley transform is a transformation of the form $Q = (I - A)(I + A)^{-1}$, where $A$ is some skew-symmetric matrix, which leads to a special orthogonal matrix $Q$. However, it is not explained how the skew-symmetric matrix $A$ is obtained, so it is unclear to me how the transform is being used. Additional details seem important for applying the method.

Experiments:
1. What are HECNN and HCNN? This should be clear from the text.
2. What is the curvature fixed to in the fixed curvature ablation?

I do believe that the proposed methods, especially the method for optimizing the curvature, are very interesting ideas that should be shared. Therefore, if the authors address most of these issues by rewriting the method section to be more coherent, consistent and complete, then I am certainly willing to increase my score.

Some more small notes:
1. There's a mistake in the definition of $\mathcal{L}^n$. I think the squared Lorentzian norms should be equal to $-K$.
2. Equation 4 is missing brackets.
3. Text below equation 5 mentions 2 and 2, which should refer to equation 2 and algorithm 2, respectively, I think.

**Questions:**

Aside from my concerns listed above, I have several questions regarding additional experiments. I think that answering these could be beneficial to the paper, although in my opinion not strictly necessary.
1. If I understand correctly, then the encoder and the decoder each have their own learnable curvature currently. What happens if each layer is given its own learnable curvature? Does performance stay approximately similar or does it increase? Do the layers learn significantly different curvatures?
2. How expensive is the moving of the parameters that is performed in algorithm 1 compared to the usual parameter update that occurs?
3. What happens when the curvature scheme is replaced by simply clipping the parameters to be within the representation radius of the new manifold?
4. In my experience, a learnable curvature is often helpful in stabilizing the loss a bit during training. Do the authors find this as well? If so, it might be nice to add a visualization of training curves somewhere (maybe the appendix).

---

> ### Author Response · Authors · 2024-11-19
>
> Thank you very much for the in-depth review. Based on your comments and shared sentiment with the other reviewers, we completely rewrote most of the methodology section and parts of the experimental setup and results sections. Hopefully the new draft is clearer and better presents our motivation and approach in designing these components.
>
> __On the Optimizer:__ We have followed your advice and replaced operations with direct notations to reduce clutter, which also helped save space for incorporating all the updates.
>
> __On Maximum Distance Rescaling__
>
> 1.	While errors are minimal under float64, its computational cost is prohibitive as most hardware is optimized for float32 operations in terms of FLOPs and efficiency. Additionally, many recent works, including the HIER experiments are moving towards float16 and float8 precisions in order to reduce memory and run time.  Given that some of the hyperbolic functions require potentially undefined operations like the cosh function, the introduced imprecisions lead to both performance degradation and instability. The HIER experiments were a good validation of our scaling layer under float16, allowing us to learn curvature and components without runtime errors.
> 2.	Aside from the empirical ablations (which showed NaN values without our modifications), there are two theoretical reasons why the existing approach falls short:
>
>     * For the hyperboloid, the projection done by only recalculates the time value. While this could be acceptable for the model parameters it alters gradient magnitudes and possibly the momentum directions due to misaligned tangent spaces, as illustrated in the new figure in our revised draft. This causes some convergence issues and training instability.
>
>    * Some of the instability arises during the update steps before projection. Specifically, curvature updates are not synchronized with manifold parameters. This means that we use hyperbolic operations that rely on the new curvature with vectors that are only defined as hyperbolic under the old curvature, including the gradients and momentums. This can cause naan and infinite values.
>
> 3.	We’ve rewritten this section to define the scaling term alone (with a visualization in the appendix) and detailed its integration with hyperbolic values. Let us know if this is still confusing so we edit it further.
> 4.	The notational inconsistencies you mentioned, along with a few others, should now be fixed. Thank you for pointing them out.
>
> __On the Convolutional Layer:__
>
> 1.	We updated the convolutional layer section to clearly explain the differences from Bdeir et al. and provide stronger motivation for its necessity.
> 2.	We have now added the definitions for Lorentz transformations in the paper appendix (due to issues with the page limit)
> 3.	We have better clarified the goal of the rotation. We mention the Stiefel manifold since we offer the option to directly learn these semi-orthogonal weights (using geoopt) in the implementation directly rather than rely on a Cayley projection.
> 4.	The problem with normal 2d convolutions is that their outputs are not necessarily hyperbolic vectors even if their inputs are. And while you could reproject the outputs after, Bdeir et al. tested this in their work and showed that it performs much worse. As for the problem with implementation by Bdeir et al., they rely on manual window patch creation followed by a lorentz linear layer. This is mathematically correct but we can no longer leverage the heavily optimized CUDA convolution implementations, significantly increasing memory and runtime costs. We now highlight these points better in the text. Our method retains CUDA compatibility by parametrizing the weights as rotation matrices before passing them into the convolution operation
> 5.	The Cayley projection we use, available in the Torch library, computes an orthogonal matrix $\textbf{Q}$ for a skew matrix $\textbf{A}$ by solving for $\textbf{Q} = (I + \textbf{A}/2)(I-\textbf{A}/2)^{-1}$. $\textbf{A}$ is probably calculated as $\textbf{A} = \textbf{U}-\textbf{U}^* +  (\textbf{V}^*)^T\textbf{V}$ where $\textbf{U}, \textbf{V} = \textbf{W}[…, :n], \textbf{W} […, n:]$, $\textbf{U}^*$ is the conjugate, and $\textbf{W}$ is the weight matrix. But we haven’t actually looked at the torch implementation.
> On Experiments:
> 5.	We now properly explain the model notations in the experimental section.
> 6.	The curvature is fixed to the literature default value of -1

---

> ### Author Response · Authors · 2024-11-19
>
> __On Questions:__
>
> 12.	This is definitely an interesting idea and we would be curious to try and add it if it is done in time.
> 13.	Seeing as the update steps are inside the optimizer are done without calculating new gradients or storing them, the mapping and transport costs are similar to inference level time and memory consumption and is negligible compared to the other operations
> 14.	Clipping parameters normally leads to information loss and non-smooth gradients which causes some training issues. Additionally, Chen et al used the sigmoid function to limit the outputs to this radius but Bdeir et al. found that it saturates too quickly which also caused performance issues. Our approach is similar to Chen in that we use a normalization function, we just provide finer control over the saturation limits and mapping slope.
> 15.	Yes actually! It helped both stabilize the loss and lead to faster convergence even if the performance gains in some cases were marginal.  We will be mentioning this in the paper and adding a graph to the appendix based on your feedback as well.

---

> > ### Comment · Reviewer_KHqp · 2024-11-26
> >
> > I thank the authors for the effort they put into the revisions. Most of my concerns have been addressed, although I think that the subsection on the Lorentz Convolutional Layer could still benefit from some more background, since the general reader might not be familiar with Lorentz boosts and rotations. Based on the revisions, I will update my score.

---

> > > ### Author Response · Authors · 2024-12-03
> > >
> > > Thank you very much for your interest and for the score adjustment. We appreciate your consideration of the revisions. As for the background, we completely agree and we did add the Lorentz Boost and Rotation explanations in the appendix now but mistakenly did not reference them in the paper yet, we would add that in the future.

---

### Official Review · Reviewer_cP51 · 2024-11-02

**Soundness:** 3
**Presentation:** 3
**Contribution:** 2
**Rating:** 5
**Confidence:** 3

**Summary:**

The paper enhances hyperbolic deep learning for computer vision by introducing new methods to improve curvature learning and computational efficiency. Key contributions include a schema for Riemannian optimizers that handle manifold changes, a Riemannian AdamW optimizer, and a maximum distance rescaling function for stability. It also presents an efficient convolutional layer to reduce memory and runtime costs. These innovations boost performance in hierarchical metric learning and classification tasks, showing robustness even with lower precision.

**Strengths:**

The paper is original, introducing innovative methods like the Riemannian AdamW optimizer and distance rescaling for better curvature learning in hyperbolic deep learning. The quality is strong, backed by comprehensive experiments that show improved performance. It is clear, with structured explanations and detailed algorithms. Its significance lies in enhancing the stability and efficiency of hyperbolic models, making them more practical for computer vision.

**Weaknesses:**

The atuhor failed to provide the results in lartge dataset, like ImageNet 1000 or 100. It will hinder the generosity of the proposed method. The implementation details is too little, like the hyperparameters.

**Questions:**

Could the author explain why only run small datasets, is the computational cost too high for large datasets?

---

> ### Author Response · Authors · 2024-11-19
>
> Thank you very much for reading the paper. We’re very glad you found it to be interesting!
>
> 1.	In terms of larger datasets, we do run the ResNet-18 models on tiny-imagenet which is a 200-class subclass of the imagenet dataset. However, the current computational requirements of hyperbolic models limit the scalability to larger datasets. The primary goal of the new convolution formulation and the bottleneck block is to reduce these computational demands while maintaining similar or improved performance. We have already achieved significant progress with some models, demonstrating a 3x reduction training time and nearly 2x reduction in memory usage (we now highlight this better in the results tables). But, further optimizations are necessary before are able to scale to larger datasets reasonably.
>
> 2.	For the hyperparameter details, we used the same setup as the Bdeir et al. experiments. We now mention this clearer in the experimental section. Additionally, we will make the code available for everyone to access after the paper is published.

---

### Official Review · Reviewer_FJpU · 2024-11-03

**Soundness:** 2
**Presentation:** 3
**Contribution:** 2
**Rating:** 3
**Confidence:** 4

**Summary:**

This paper extends the AdamW optimizer to Riemannian manifolds and proposes a method for curvature optimization and a Riemannian convolutional layer. Despite extensive experiments, the approach lacks theoretical insights, such as the rationality of the proposed several blocks.

**Strengths:**

1. Extension of several existing Lorentz blocks to Riemannian Setting.

**Weaknesses:**

1. Generalizing Adam into AdamW is straightforward. This can be readily done by the Riemannian operators at hand.

2. In Riemannian optimization, usually use retractions and vector transport instead of exp and Levi-Civita transport. This has become a common technique. The reason why the authors focus on the expensive exp and LC trans is not clear.
3. As for the mentioned Kim et al. (2023), it is more generally called trivilization. One of the biggest benefits is the direct application of Euclidean optimizer. So, the missing trivialized AdamW makes this work incomplete.
4. The key ingredients in L170-185 for the curvature optimization are confusing. We can post-optimize K and transport every data into the new space via tangent space. so, why can we not do prior-optimize K and post-transport similarly, by tangent space? A more important question is why should we bother to transport data, esp manifold data. Why can we not view the manifold data unchanged? As doing the transformation by single tangent space is clearly just an expedient, just as no transformation. if the authors want to show the benefits of post-K-and-transformation, the authors should show some theoretical analysis of which properties have been maintained or respected by the proposed methods.
5. The motivation and benefits of the convolutional layer are unclear. Except for ensuring the output as the L space, which geometric properties does the convolutional layer respect? Does the proposed conv layer have properties such as equivariance [2]?
6. Experiments should be done on large datasets, as the proposed method is targeted to the CV and most of the experiments are on small datasets.
7. Finally, does the proposed RAdamW and curvature optimization have a better theoretical guarantee than the existing ones?

Other minor comments:
Some notations lack clarity or rigorousness and expressions lack motivation:

- $v_i$​​ in L centroid should lie in simplex
- L 216-219,  what does the L case mean? does it mean the centroid between $\theta ^{t-1}$ and $\bar{0}$​​? why is it formulated in this way is not clear.
- $x_t$ in L 129.
- the notation for the Stiefel manifold is quite exclusive.
- what is LorentzBoost in Eq. (5)
- why centroid is $(\sqrt{K}, 0)$. Does it come from the hom space or mimicking some aspect of the Euc space?

**Questions:**

1. in the experiments, is the K identical across layers or different depending on the layer?



1. Trivializations for Gradient-Based Optimization on Manifolds
2. ManifoldNet: A Deep Neural Network for Manifold-Valued Data With Applications

---

> ### Author Response · Authors · 2024-11-19
>
> Thank you for the time and effort you put into reading and reviewing the paper.
>
> Just as a main point before going into the discussion, we would like to make clear any misconception about hyperbolic vectors on the hyperboloid. When describing points in the Lorentz model, we call the first dimension the time component $x_t$ and the remaining dimensions the space components $\textbf{x}_s$, such that $\textbf{x}\in\mathbb{L}^n_K=[x_t, \textbf{x}_s]^T$.
>
> Given that, $\langle \textbf{x}, \textbf{x}\rangle_{\mathcal{L}}= \frac{-1}{K}$ and $\langle \textbf{x}, \textbf{y}\rangle_{\mathcal{L}} := -x_ty_t+\textbf{x}_s^T\textbf{y}_s$, we get $x_t = \sqrt{||\textbf{x}_s||^2+1/K}$. As such, the origin, where all the space values are equal to zero, becomes $\overline{\textbf{0}} = [\sqrt{1/K}, 0, \cdots, 0]^T$ and vectors cannot be directly interchanged between manifolds of different curvatures since the time value is reliant on the curvature.
>
> As for the comments:
>
> 1.	Although generalizing from RAdam to RAdamW is conceptually straightforward, it has not yet been addressed in the existing literature. By sharing our reasoning and implementation, we aim to establish a baseline and provide an accessible solution for future research that requires it.
>
> 2.	Retractions and vector transport are actually used in the model, particularly in the residual connections, initial image projection, and activation functions. However, certain components, such as batch normalization, have shown empirical instability. The approximation errors introduced by retractions and vector transports can often result in invalid values. Additionally, we use the exact operations (exponential map and parallel transport) in the optimizers as they provide greater accuracy and do not impact the inference requirements. Furthermore, since gradients are neither stored nor calculated during optimizer update steps, the computational cost of these exact operations is significantly reduced during training.
>
> 3.	As you mentioned, trivialization enables the use of Euclidean optimizers, making a trivialized AdamW unnecessary. However, a significant drawback of trivializations in some works is the need to manually handle the conversion between Euclidean and Riemannian gradients for each component, including writing a custom backpropagation function for these classes. To address this, we opted for a non-trivialized approach that only requires initializing the weights as manifold parameters, allowing the Riemannian optimizers to handle these conversions automatically.
>
> 4.	Thank you for raising this point. Your feedback prompted us to revise parts of the methodology section, including the curvature learning approach, and to include additional visual aids for clarity. You are correct that the default implementation of GeoOpt leaves the manifold parameters unchanged, which caused highly unstable training. This instability arose because some update steps incorrectly treated parameters as belonging to the new manifold. As a result, the time values of the unchanged parameters became invalid, leading to NaN or infinite values when used with the new curvature operations that depend on a correct time value.
> Simply recalculating the new time values without proper mapping also introduces issues, as it alters the gradient magnitudes and the momentum directions. To address this, we chose to perform mapping  and parallel transport onto the tangent plane of the origin then back onto the new manifold. This approach preserves both the magnitudes and relative directions of parameters, as well as their gradients and momentum. We have detailed this process, along with its theoretical basis, in the revised draft.
>
> * Specifically, we used the tangent plane of the origin because it remains parallel to the tangent spaces after curvature updates, enabling simpler vector projection and allowing direct reuse of origin tangent mappings without modification.
>
> * Finally, you are correct that it is not strictly necessary to update the curvature last. As long as we track both the old and new curvatures, update parameters based on the old curvature, and then map them onto the new manifold, the process remains valid.
>
> 5.	The proposed convolutional layer ensures that any transformation is strictly a rotation, preserving the norms of vectors while altering only their orientations. This guarantees that the output remains within the Lorentz space (since the norm of the space values remain unchanged which does not require a change in the time component), aligning with the definition of a hyperbolic operation as mentioned by Bdeir et al. and prior research. To achieve a complete hyperbolic transformation, a boost operation can then be applied, modifying the norms while keeping the orientations unchanged.

---

> ### Author Response · Authors · 2024-11-19
>
> 6.	We have updated the results to include the computational requirements of the models for CIFAR-100. Currently, these requirements limit the scalability to larger datasets. However, the primary goal of the new convolution formulation and the bottleneck block  is to reduce these computational demands while maintaining similar or improved performance. We have already achieved significant progress with the ResNet-50 model, demonstrating a 3x reduction in training time and nearly 2x reduction in memory usage, enabling training with reasonable VRAM and time constraints. That said, further optimizations are necessary before scaling to larger datasets becomes feasible.
>
> 7.	As we demonstrate, the previous curvature optimization method is fundamentally flawed. Furthermore, to the best of our knowledge, no implementation of RAdamW has been developed so far. Therefore, we believe the proposed components represent meaningful improvements over the existing approaches.
>
> 8.	Many of your comments regarding the annotations have been addressed—thank you for your detailed feedback. We would just note that the L case is the annotation we used for the Lorentz manifold, and is defined in the background section. Additionally, we have included the mathematical definitions of the Lorentz boost and Lorentz rotation in the appendix, as requested by other reviewers as well.
>
>
> 9.	The encoder and decoder are designed to have independently learnable curvatures, allowing for flexibility in their respective manifold representations. However, all blocks within the encoder share the same curvature. Exploring the use of different curvatures for individual blocks within the encoder presents an intriguing direction for future research.

---

> ### Comment · Reviewer_FJpU · 2024-11-22
>
> Thanks for the authors' reply.
>
> Why do you say that trivialization will involve manual backpropagation? Trivialization should be much simpler, as it uses Riemannian exp or other surjective to parameterize the manifold parameters. I know and have experimented with several geometries that it might be better. In hyperbolic space, the Riemannian exp can be auto-graded by torch. I know manual BP might be necessary for a few computations in matrix manifolds. But vector manifolds, like the hyperbolic, are much simpler, as Riemannian computations are mostly decomposition-free. Currently, I do not see any operation that cannot be auto-differentiated.
>
> Besides, even if you need manual BP, it is quite easy. This is a quite basic exercise in geometry. I do not think this is a hard issue.
>
> In many cases, together with trivialization, the final expression can be further simplified, such as HNN++ and ones on matrix manifolds:
>
> Matrix Manifold Neural Networks++
>
> Building neural networks on matrix manifolds: A Gyrovector space approach
>
> The Gyro-structure of some matrix manifolds

---

> ### Comment · Reviewer_FJpU · 2024-11-22
>
> Another issue concerns the BN layer.
>
> I could not find the explicit formulation of the BN layer in your work. Do you follow Eq. (6) in [1]? Can BN normalize sample statistics? As noted in [2], the combination of exp-log-pt operations cannot normalize the sample distribution, which is crucial for BN. Only in certain specific cases, such as on the SPD manifold, where it coincides with the GL(n)-action, can it achieve normalization [3].
>
> [1] Fully Hyperbolic Convolutional Neural Networks for Computer Vision
>
> [2] Manifoldnorm: Extending normalizations on Riemannian manifolds
>
> [3] Riemannian batch normalization for SPD neural networks

---

> ### Author Response · Authors · 2024-11-22
>
> Thank you for going over the rebuttal!
>
> We might have misspoken a bit, by manual backpropagation we did not mean explicitly calculating gradients by hand. As you pointed out, the operations are definitely auto-differentiable. However, when optimizing these parameters, the auto-computed Euclidean gradients must be transformed into Riemannian gradients. While this transformation is very straightforward, it must be explicitly defined and applied before the optimization step.
>
> As examples of this:
>
> * [1] and [2] Perform this during the ToPoincare projection operation that creates a class for the parameter that projects the gradient into Riemannian everytime.
> * [3] and [5] Iterate over all the parameters before optimization, check if the variable parameterizes a hyperbolic operation, then cast the gradient accordingly.
> * [4] Creates two different optimizers, during operation definition, it adds the trivialized hyperbolic parameters to a list and then optimizes that list separately with a Riemannian optimizer.
>
> Other approaches also modify the automatic backdrop function for every hyperbolic module to perform the gradient casting.
>
> While the process is theoretically very straightforward, the literature varies significantly on how and where to perform these gradient updates. We chose a centralized approach that minimizes the effort for future researchers to adopt.
>
> Additionally, we did consider using trivializations, which is a valid method we've employed in other hyperbolic research. We also found that the existing code for [6] includes a trivialized batch normalization with cast Euclidean parameters. However, in our implementation, this approach yielded worse empirical results (in our models at least). It remains a very interesting direction to explore trivializations for the entire model, as some researchers report improvements, particularly when learning curvature (although these improvements might also be attributed to the poor update scheme in this particular case).
>
> While we encourage dedicated research to compare the two possible parameterization approaches directly, our work here at least aimed to support the literature on direct hyperbolic parameter learning, so it can also benefit from stable curvature learning.
>
> We even have the option in the library and the paper to choose between a trivialised Cayley projection approach for the convolutions or learn the rotation matrices directly on the stiefel manifold. However we did not see significant performance differences between the two.
>
>
> ---
>
> [1] Ermolov, Aleksandr, et al. "Hyperbolic vision transformers: Combining improvements in metric learning." Proceedings of the IEEE/CVF Conference on Computer Vision and Pattern Recognition. 2022.
>
> [2] Kim, Sungyeon, Boseung Jeong, and Suha Kwak. "Hier: Metric learning beyond class labels via hierarchical regularization." Proceedings of the IEEE/CVF Conference on Computer Vision and Pattern Recognition.
>
> [3] Atigh, Mina Ghadimi, et al. "Hyperbolic image segmentation." Proceedings of the IEEE/CVF conference on computer vision and pattern recognition. 2022.
>
> [4] Dai, Jindou, et al. "A hyperbolic-to-hyperbolic graph convolutional network." Proceedings of the IEEE/CVF conference on computer vision and pattern recognition. 2021.
>
> [5]  Ganea, Octavian, Gary Bécigneul, and Thomas Hofmann. "Hyperbolic neural networks." Advances in neural information processing systems 31 (2018).
>
> [6] Bdeir, Ahmad, Kristian Schwethelm, and Niels Landwehr. "Fully Hyperbolic Convolutional Neural Networks for Computer Vision." arXiv preprint arXiv:2303.15919 (2023).

---

> ### Author Response · Authors · 2024-11-22
>
> The model we based our work on is the one defined in [1] including the batchnorm layer. We understand that it is based on the method by [2] but replaces the frechet mean with the centroid for faster calculation. Additionally they provided a theoretical decomposition for the batch norm as a recentering and rescaling approach, and provide the operations for each (recentering using centroid and parallel transport, and rescaling on the tangent of the origin).
>
> However, we were not aware of [3], although our paper does not target the batchnorm, that is very helpful insight for future work or at least to mention in this one to encourage alternatives. From a quick view, the proposed BN seems similar to that of algorithm 1 in [3], we'll take a deeper look and perform some tests to see if the BN from [1] can normalize sample statistics.
>
> [1] Bdeir, Ahmad, Kristian Schwethelm, and Niels Landwehr. "Fully Hyperbolic Convolutional Neural Networks for Computer Vision." ICLR 2024.
>
> [2] Lou, Aaron, et al. "Differentiating through the fréchet mean." International conference on machine learning. PMLR, 2020.
>
> [3] Riemannian batch normalization for SPD neural networks

---

> ### Comment · Reviewer_FJpU · 2024-11-23
>
> I thank the authors for the further clarification.
>
> - For trivialization. Only with empirical results, can we safely say it is not working well on hyperbolic against the Riemannian counterparts. The geometries that I experimented on tell me in several cases, it is easier and better, as it can use lots of optimizers to tune, without the efforts to re-code the Euclidean ones following the Riemannian rule.
>
> - For the BN issues. The Riemannian BN in Lou is a natural variant of SPDNetBN. It is thoroughly discussed in ManifoldNorm. The key issue of log-exp-pt BN lies in its incapability of normalizing sample statistics in general, which was discussed in ManifoldNorm several years ago (see the homogeneous case Algs. 1-2).  Without the ability to control sample statistics, it is simply a transformation, which looks like BN but fundamentally does different things. This is an interesting and complex part of geometries.
>
> ***
> My main concern is that this work lacks in-depth insight in quite a few ways. It is more like a stack of existing techniques, although the author made efforts on several aspects. Therefore, I will keep my original score.

---

> > ### Author Response · Authors · 2024-11-24
> >
> > Thank you for taking the time to discuss this further.
> >
> > * For trivialization, we have again tried the batchnorm with the trivialized approach, and given that the rotation convolution is based on a Cayley projection and the boost and Lorentz MLR are already Euclidean parametrized, that would make the model (for classification with ResNet not HIER or VAE), fully trivialized. In this scenario, we did see worse results (80.11% vs 80.86% for Resnet-50 in CIFAR-100 averaged over 5 runs). We could include that in the ablations if you think it benefits the paper. We appreciate your expertise in the matter as we have mainly modeled on the Lorentz manifold previously, and have debated either approach since both seemed popular. In most of our cases, extensive modeling with trivialization showed that a mix of both approaches seemed to benefit the most, although this again could be limited to the modeling on the hyperboloid.
> >
> > * In terms of the bach norm, this definitely piqued our interest and we will be looking at it going forward, especially because intuitively both approaches seem very similar as you mentioned. We just hope that this did not affect your view of our paper since we do not tackle the batchnorm at all here.
> >
> > ---
> >
> > We respect your decision to keep the scoring as is although we somewhat disagree with the framing that it is a stacking of existing techniques. Especially because the paper aimed to optimize and stabilize existing well-published frameworks and approaches which we believe has merit in itself. In addition to some entirely different approaches to address existing issues. These problems are not rooted as much in the actual geometry itself but rather in the approaches and computational limitations and so they require less theoretical solutions. However, this rebuttal discussion was very enjoyable for us as it held a lot of learning merit, so we thank you again.

---

> ### Comment · Reviewer_FJpU · 2024-11-24
>
> I may have come across a bit strongly; my intention was merely to discuss the underlying nuances of several related issues, which I believe could be important factors. Personally, I value more in-depth insights and encourage the authors to delve deeper into this line of research.
>
> Thank you for your efforts.

---

### Official Review · Reviewer_CXX3 · 2024-11-03

**Soundness:** 1
**Presentation:** 2
**Contribution:** 1
**Rating:** 3
**Confidence:** 3

**Summary:**

In this paper, the authors propose several methods to improve learning in hyperbolic space: a more principled way to learn the curvature as a parameter, a Riemannian version of AdamW, a maximum distance rescaling function that allows the usage of fp16 precision, and a more efficient hyperbolic convolutional layer. Experiments are conducted on the hierarchical metric learning problem, image classification, and image generation with a VAE architecture.

**Strengths:**

Proposes a variety of new techniques to improve hyperbolic learning, including a Riemannian version of AdamW, a more principled curvature learning algorithm, a max rescaling algorithm for better numerical stability, and new efficient hyperbolic convolution.

**Weaknesses:**

The use of the proposed Riemannian AdamW algorithm is not well-supported, as it is only used in the hierarchical metric learning experiment, there is no ablation on using Riemannian AdamW, and there is no theoretical justification.

It isn't clear whether the proposed Lorentz-Core Bottleneck Block was used for any of the experiments and no ablation study is done on this component.

Improvements using this method either barely improves is comparable, or is slightly worse than the proposed baselines in hierarchical metric learning task and classification tasks (Tables 1, 2, 3).

Minor point: In Table 1, some of the numbers for the proposed method (LHIER) are bolded despite not being better than the baseline numbers.

**Questions:**

1. Which experiments was the Lorentz-Core Bottleneck Block used in? And are there ablations showing the effectiveness of the proposed block?
2. Is it possible to compare Riemannian AdamW to Riemannian Adam or Riemannian SGD?

---

> ### Author Response · Authors · 2024-11-19
>
> Thank you for the time you put into reading and reviewing the paper.
>
> You are correct on a lot of the points you make so to answer your questions:
>
> 1.	The experimental setup section lacked clarity, particularly regarding where the core-bottleneck block was applied. We have rewritten this section to explicitly state that the core-bottleneck block is used in the ResNet-50 HECNN+ model. In our implementation, we replace all the alternating Euclidean and hyperbolic bottlenecks from Bdeir et al. with the Lorentz core block and directly compare their performance and computational costs. These details are now emphasized in the ResNet-50 table, alongside the computation savings achieved.
>
> 2.	AdamW has become a standard optimizer for many vision tasks, and the HIER paper uses AdamW for its experiments. To ensure fair comparisons, we also adopted AdamW in our study. Retraining HIER with Adam or comparing results obtained with different optimizers would not provide an ideal benchmark. Moreover, since the differences between Adam and AdamW are relatively minor, we adapted AdamW for hyperbolic learning. The revised paper now includes a more detailed explanation of the theoretical reasoning behind these changes. Additionally, to provide comprehensive comparisons, we reran the ResNet HIER experiments with Adam, AdamW, and SGD, and we have included these results here:
>
> |        |         | CUB  |         |         | Cars |         |         | SOP  |        |
> | ------ | ---- | ---- | ---- |---- |---- |---- |---- |---- |---- |
> |        | R@1  | R@2  | R@3  | R@1 | R@2 | R@3 | R@1 | R@2 | R@3 |
> | RSGD   | 64.2 | 72.1 | 78.6 | 73.9 | 81.4 | 84.8 | 67.8 | 73.9 | 80.1 |
> | RAdam  | 70.1 | 79.8 | 86.9 | 87.6 | 92.0 | 94.9 | 79.8 | 90.3 | 95.2 |
> | RAdamW | 72.4 | 81.5 | 88.4 | 89.1 | 93.5 | 96.1 | 81.3 | 92.1 | 96.8 |
>
> These results are based on only 2 run averages but we will update them and add them to the appendix as soon as the experiments are done.
>
> 3.	Finally, while some empirical accuracy improvements are modest, we have revised the experiment section to better highlight the overall performance/efficiency gains that a lot of our components aim for. We mainly focus on maintaining or improving accuracy while significantly reducing computational requirements, as well as enabling stable curvature learning without crashes or divergence during training.

---

> > ### Comment · Reviewer_CXX3 · 2024-11-27
> >
> > Thank you for your efforts in writing the rebuttal.
> >
> > I still have reservations on the justification of some of the proposed contributions. While the ablation study demonstrated the superiority of Riemannian AdamW to Riemannian SGD and Adam, Riemannian AdamW is still only being used on the HIER experiment. Combined with the lack of theoretical justification, I feel that the justification is insufficiently robust. Similarly, the Lorentz block is only investigated with one comparison, with only comparable results. It is more efficient, but it is not clear if the performance will be better or comparable in other settings.

---

> > > ### Author Response · Authors · 2024-12-03
> > >
> > > Thank you for revisiting the rebuttal and the paper. We acknowledge that including numerous components constrained the depth of analysis and experiments we could perform given the limitations and the already existing need for multiple experimental settings. However, we understand that a more in-depth exploration would strengthen the argument for the adoption of approaches.

---

### Author Response · Authors · 2024-11-19

Thank you for your thoughtful review! We greatly appreciate your feedback and have made significant revisions to improve the paper's presentation. In particular, we have reworked the methodology, experiment, and results sections to address concerns about the clarity and theoretical grounding of some proposed components and experimental setups. We have now included detailed explanations for these components, refined the experimental setups, and added additional visualizations to enhance the understanding of key concepts. We also now highlight better the main objectives of the paper with regards to computational efficiency and training stability while maintaining or improving the accuracies.

---

### Meta-Review · Area_Chair_BZYg · 2024-12-20

**Metareview:**

This paper proposes a method for learning the curvature of a hyperbolic space using Lorentz geometry. The authors highlight the instabilities of previous approaches to curvature learning, develop a new algorithm leveraging the properties of the tangent plane at the origin, and also develop a method to performing convolution in Lorentz spaces. Empirically, the proposed algorithm is integrated into various baselines across several vision tasks, such as metric learning and image generation.

The primary concern raised by reviewers is the contribution of the work. While the approach has merits, it resulted in only marginal improvements in certain cases. Additionally, some ideas, such as the Riemannian ADAM optimizer, are well-studied and not novel. The discussion following the author-reviewer interaction and further deliberations among the reviewers and the AC led to a consensus that, despite its merits, the work is not yet ready for publication. As such, the AC, sadly recommends `rejection`.

We hope this review helps the authors identify areas for improvement. On a related note, the use of the Stiefel manifold for achieving rotation, as proposed for Lorentz convolution, is not entirely adequate. While the columns of a point on the Stiefel manifold are orthonormal, they do not inherently represent rotation. Instead, the Stiefel manifold can be seen as a linear mapping with an additional orthogonality constraint.

**Additional Comments On Reviewer Discussion:**

The paper was reviewed by five experts in the field and initially received one positive score and four negative ones. Discussions during the author-reviewer period focused on clarifying the contributions of the paper (e.g., the Riemannian AdamW optimizer) and positioning it in terms of novelty relative to prior work. While the authors responded to the raised concerns, none of the reviewers championed the paper, as its theoretical contributions were not perceived to be significant.  The AC agrees with the reviewers' main concerns regarding the technical novelty of the work and, despite its merits, regretfully recommends rejection.

---

### Decision · Program_Chairs · 2025-01-22

Reject